# Deformation Estimation of Textureless Objects from a Single Image

**DOI:** 10.3390/s24144707

**Published:** 2024-07-20

**Authors:** Sahand Eivazi Adli, Joshua K. Pickard, Ganyun Sun, Rickey Dubay

**Affiliations:** 1Department of Mechanical Engineering, University of New Brunswick, 15 Dineen Drive, Fredericton, NB E3B 5A3, Canada; dubayr@unb.ca; 2Eigen Innovations Inc., Fredericton, NB E3B 1S1, Canada; josh.pickard@eigen.io (J.K.P.); grace.sun@eigen.io (G.S.)

**Keywords:** deformation estimation, image dataset, textureless deformed object, single image, graph convolution, label generation

## Abstract

Deformations introduced during the production of plastic components degrade the accuracy of their 3D geometric information, a critical aspect of object inspection processes. This phenomenon is prevalent among primary plastic products from manufacturers. This work proposes a solution for the deformation estimation of textureless plastic objects using only a single RGB image. This solution encompasses a unique image dataset of five deformed parts, a novel method for generating mesh labels, sequential deformation, and a training model based on graph convolution. The proposed sequential deformation method outperforms the prevalent chamfer distance algorithm in generating precise mesh labels. The training model projects object vertices into features extracted from the input image, and then, predicts vertex location offsets based on the projected features. The predicted meshes using these offsets achieve a sub-millimeter accuracy on synthetic images and approximately 2.0 mm on real images.

## 1. Introduction

Computer vision plays a crucial role in various industries, particularly in the automated inspection of industrial objects. While numerous methods and techniques have been developed for analyzing 2D information extracted from images (e.g., edges or counters), less emphasis has been placed on addressing imprecise 3D information extracted from objects, e.g., object’s dimensions. Given the importance of 3D information in computer vision applications, such as mapping 2D information onto a corresponding 3D model of the object, understanding sources of imprecision in 3D information is imperative. Unforeseen fluctuations in environmental conditions during the manufacturing process are one of the primary contributors to these inaccuracies by generating undesired deformations on the produced objects. Hence, it is imperative to explore methods that enhance the precision of 3D information pertaining to manufactured objects by estimating these deformations.

This study introduces a novel method to estimate deformation in objects using a single RGB image. It treats the manufacturing process as a black box and focuses on the impact of deformations on the accuracy of objects’ dimensions. Given the prevalence of geometric variations or deformations among plastic industrial components, which are often devoid of texture, this paper primarily focuses on estimating the deformations of marginally deformed textureless plastic objects. This approach aims to assist manufacturers in improving the quality of their inspection processes, as the mapping process is an integral component of object inspection.

Marginally deformed textureless plastic objects encompass a significant portion of primary plastic products manufactured by producers. These products typically exhibit a uniform color and lack any alphanumeric characters on their surfaces, hence being described as textureless. While the proposed method may also apply to textured objects, other techniques such as SIFT [1] or SURF [2] can be utilized to harness the valuable 2D information provided by texture.

The term ‘marginal’ distinguishes between the large, elastic deformations common in computer vision tasks like object detection, and the plastic, relatively minor deformations occurring during manufacturing. While the proposed solution is invariant to the material of the object, the focus is primarily on components crafted from plastic, as marginal deformations are more frequently observed in them.

This paper presents the object’s model as a mesh consisting of vertices and faces. In this context, deformation refers to the displacement of the deformed model’s vertices from their original positions in the undeformed model, measured using the Euclidean distance between two 3D positions. During deformation, the order of the model’s faces remains constant. According to this definition, the paper proposes a training model to estimate the offsets or displacements of each vertex in the deformed model using a computer-aided design (CAD) model of the undeformed object and a single RGB image of the deformed object. Since industrial practice often relies on inspecting objects from a single image, this method leverages a single image to enhance the proposed model’s practicality. This proposed solution poses some challenges that require careful consideration and resolution.

Dataset generation presents an initial challenge as, to the best of our knowledge, no suitable dataset exists that meets our specific requirements. Therefore, we have created an exclusive dataset of five different textureless objects. This dataset includes their undeformed CAD models, numerous deformed versions of these models, corresponding images of deformed models from various viewpoints, and the pose of the objects with respect to the camera. This contribution can benefit researchers working on deformed textureless models, as well as those interested in textureless object pose estimation.

Label generation poses a significant challenge, as synthetically deformed objects exhibit variations in the number and order of vertices compared to the initial model. This results in multiple potential solutions for representing deformed objects with a high number of vertices using the initial mesh with a limited number of vertices. This occurs because for a single vertex in the initial mesh there can be multiple potential correspondences in the deformed object. To address this issue and determine the optimal solution, involving the identification of the best corresponding vertex in the deformed model for each vertex in the initial model, a label generation method is devised to seamlessly deform the initial model. This important contribution is beneficial for researchers in the 3D object reconstruction field, allowing the creation of optimal labels.

The third challenge is to design a training model tasked with estimating offsets of the deformed model’s vertices based on a single RGB image of the deformed object and the camera’s intrinsic and extrinsic parameters. Drawing inspiration from Pixel2Mesh [3], a model is formulated to establish a connection between extracted features from the input image and corresponding generated labels.

The structure of the paper is outlined as follows: the background of the research is provided in Section 2, and the methodology employed in this study is detailed in Section 3. Section 4 is dedicated to describing the experiment setup and presenting the results. The paper concludes with a summary in Section 5.

## 2. Related Works

To the best of our knowledge, the most relevant solution to the problem of object deformation estimation from a single image of the deformed part and an initial undeformed mesh model is presented through 3D object reconstruction based on a single image of the object. Thus, it is crucial to review the current methodologies within this domain. The primary objective of 3D reconstruction methods is to recover the 3D structure and geometry of an object from one or more images. This problem finds applications in various fields such as robotics [4,5,6], computer vision [7,8] and industry [9,10,11]. In the past, geometric methods were employed to tackle this problem, which relied on mathematical or geometric solutions that necessitated the use of multiple well-calibrated cameras to capture images of the object. Stereo-based methods [12] followed a similar approach, but they involved capturing images of the object from different angles and employing the triangulation principle to address the 3D reconstruction problem. Traditional methods relied on precisely calibrated cameras and multiple images to provide adequate information for accurate 3D reconstruction of objects. However, relying on a single image presents several challenges, such as the loss of 3D information and the need for large datasets, as described in [13]. Recent deep learning approaches [14,15,16] have been developed to address these challenges and achieve 3D reconstruction using a single image.

In deep-learning-based methods, three main approaches are employed to depict 3D shapes [17]: volumetric (Section 2.1), point clouds (Section 2.2), and mesh (Section 2.3). Each representation is explored in different sections, with a particular focus on elucidating the rationale behind selecting the mesh representation for this research.

### 2.1. Volumetric Representation

Volumetric representation is a method used to represent three-dimensional shapes through a grid of volumetric pixels, also known as voxels. The primary drawback of this approach is the significant computational complexity, requiring a large amount of memory for storage and computation. This issue is compounded by higher resolutions requiring even more memory, restricting the practicality of utilizing this representation for training models. Thus, the primary challenge in volumetric representation is to achieve a high-resolution representation while keeping computation costs reasonable.

One promising solution is the coarse-to-fine approach, also known as multi-staging. This method begins with a low-resolution representation and gradually increases the resolution through up-sampling. For example, Yang et al. [18] utilized two up-convolutional layers to achieve a high-resolution voxel grid; and in [19], the same grid is constructed slice-by-slice using a long-term recurrent convolutional network (LRCN). Additionally, local region refinement was introduced in [20], where an additional CNN module refined the coarse grid resolution. Despite the ease with which deep learning architectures can leverage volumetric representation, the requirement for high memory and additional post-refining processes to increase resolution make it computationally expensive.

### 2.2. Point Cloud Representation

A three-dimensional (3D) shape can be represented by an unordered set of 3D points. While this representation is efficient in terms of memory usage, fitting such unordered point clouds into convolutional architecture poses a challenge. To address this challenge, three different solutions have been proposed in the literature. In [21], point clouds were treated as a matrix; whereas in [22], 3D points were considered as one or multiple three-channel grids; and finally, depth maps from multiple viewpoints were used in [23]. The last two solutions are computationally efficient as they use grid representations that can be easily fed into convolutional networks, while the first representation requires an additional fusion step. This representation has certain limitations, including large data sizes, the incapacity to capture texture information of the object, and the introduction of noise due to sensor limitations. A point cloud decoder was introduced in [24] to generate a coarse 3D point cloud based on extracted image features to address the sensor limitations. Then, they used a 3D Gaussian splatting technique to reconstruct the object in 3D. While this technique requires less memory to store a 3D object than point cloud representation and can provide detailed texture information, it fails to capture fine features and sharp edges.

### 2.3. Mesh Representation

Numerous methods employing a mesh representation of a 3D shape estimate a deformation field to reconstruct the shape. Several different deformation models exist, including vertex deformation, free-form deformation (FFD), and procedural models. Vertex deformation estimates the linear displacement of each vertex and requires one-to-one correspondence between the vertices of the input template and those of the output 3D model. FFD deforms the space around the template instead of its vertices. A recent study [25] explored using procedural model parameters to deform an initial mesh. This method utilizes a differentiable renderer to iteratively adjust the parameters of a procedural generator, gradually deforming the mesh to best match the input image.

This paper utilizes vertex deformation to create a new 3D model. This method offers more precise control over the shape compared to free-form deformation (FFD), which can sometimes cause unwanted distortions. Additionally, vertex deformation provides more flexibility than procedural models, which can struggle with intricate details and are limited by their pre-defined rules.

Reference [26] introduces DeformNet, which uses an up-convolutional network to decode the high-level representation of the input image into the FFD field. In a similar approach [27], the final 3D shape resulting from applying the predicted FFD field to the initial mesh model was further refined by adding a residual defined as a weighted sum of all the 3D models retrieved from the database. Multiple templates were deformed using FFD, and the one with the highest accuracy was selected in [28].

In most methods of deformation-based 3D reconstruction, deep neural networks learn the deformation field during the training process and deform the initial 3D model during the test stage. The encoder–decoder architecture is commonly utilized in these methods, where the convolutional operation encodes information, and a fully connected layer decodes information from the latent space. For example, in [29], two fully connected layers estimate the deformation field, which is then applied to a sphere as an initial 3D model. Reference [3] proposed a model that relies on multiple layers of graph convolutional networks (GCNs) to deform the initial mesh model (an ellipsoid with a low count of vertices) based on information extracted from an input image from a convolutional neural network (CNN). A significant drawback of this approach is its failure to evaluate potential solutions for fitting the initial ellipsoid to labels with a high count of vertices. This raises a question regarding whether the resulting 3D model is the most accurate representation achievable for the target model using the initial ellipsoid. We aim to address this question through the label generation pipeline.

In conclusion, mesh representation proves to be easily manipulable through the utilization of GCNs and is more computationally efficient compared to both voxel-based and point cloud-based representations. Consequently, this paper employs mesh representation to offer a solution to the existing deformation estimation problem by introducing a unique label-generation method to accurately generate mesh labels of the deformed models.

## 3. Method

In this section, we elaborate on the three primary components of this research: dataset generation (Section 3.1) provides detailed information regarding the creation of an exclusive dataset of deformed objects. Label generation (Section 3.2) is dedicated to explaining the proposed algorithm for sequentially deforming the mesh to generate labels, while model design (Section 3.3) explores the specifics of the model designed for predicting deformed meshes.

### 3.1. Dataset Generation

The dataset plays a pivotal role in training a model effectively. In this particular scenario, the dataset must encompass the following elements: the initial undeformed mesh model of the object in OBJ format, deformed versions of the initial model (Section 3.1.1), RGB images capturing the deformed objects from various viewpoints, and the pose of the deformed objects in relation to the camera (Section 3.1.2). We sought to evaluate the performance of the proposed training model on real-world data. To achieve this, we created a focused dataset consisting of two physical objects captured from various viewpoints using a limited number of images (Section 3.1.3).

This study investigates the performance of the proposed method on five different textureless plastic components: skateboard, bracket, round flat receptacle lid (RFRL), oval flange, and vent cover (see Figure 1). While the objects vary in shape and geometry to ensure comprehensive evaluation across diverse geometries, they all share the characteristic of being thin, facilitating their deformation. In Figure 1, the undeformed versions or initial models of the studied objects are depicted. These objects have different numbers of vertices and faces, as described in Table 1. Since most industrial products come with readily available initial CAD models in OBJ format from their manufacturers, we assume access to such models for this study. As previously mentioned, these initial models within the dataset reflect the real-world industrial components by having varying numbers of vertices and faces.

#### 3.1.1. Deformed Object Generation

To obtain images of the deformed parts, the initial undeformed models must first be synthetically deformed using the Ansys (version 2024 R1) software. Leveraging Ansys simulations allows for the replication of deformations closely resembling those occurring during the manufacturing process. Initially, the undeformed mesh models are solidified using Autodesk Fusion 360 (version 2.0.18961) software. The solid objects, in STEP format, are then imported into Ansys to simulate the application of static forces. We chose polypropylene (PP) as the material for all objects in the simulation. Polypropylene is a widely used material for many industrial plastic components due to its properties, e.g., strength, lightweight, and chemical resistance. Table 2 summarizes the applied forces (range and steps) and the resulting deformations (range and steps) for all studied objects.

All forces applied to the objects in Table 2 act in the *Z* direction of the object coordinate systems. The magnitudes are chosen to induce clearly visible and easily recognizable deformations with the naked eye. This approach allows us to evaluate the proposed training model’s performance in handling various deformation severities in both the positive and negative *Z* directions. The specific deformation directions are carefully selected to represent the most common types encountered during the industrial manufacturing of plastic parts. The rationale for selecting these deformation step values is addressed in the label generation section (Section 3.2). In this work, we focus on the magnitude of the maximum deformation (MD), not its direction. A negative sign indicates that the primary deformation occurs in the negative *Z* direction of the object’s coordinate system. Figure 2, Figure 3, Figure 4, Figure 5 and Figure 6 illustrate example deformed states for the skateboard, bracket, RFRL, oval flange, and vent cover, respectively. These objects were obtained through static force simulations conducted in the Ansys software. For illustrative purposes, we selected objects that experienced the highest as well as the lowest deformations in both the positive and negative directions of the *Z* axis, among the deformed models leveraged for training the designed model. Each image uses a red–blue color map, where red indicates the maximum deformation and blue indicates minimum or no deformation. The applied force and the maximum deformation of each object are labeled within the figure caption as *F* and MD, respectively.

Figure 2, Figure 3, Figure 4, Figure 5 and Figure 6 depict screenshots captured from the Ansys static force simulation work-space. Each image showcases a deformed object meticulously positioned to illustrate the experienced deformation clearly. For reference, a black wireframe representing the undeformed model is included in each figure. This visual aid allows for easy comparison and highlights the magnitude and direction of the deformation experienced by the object.

#### 3.1.2. Image Rendering Process

The Blender (version 3.3.6) software was used to generate rendered images of the deformed objects under controlled conditions. These images maintained consistent scene and camera properties across all objects, as detailed in Table 3.

For efficient rendering, each object’s size was scaled down to a manageable range of [−10.0 mm,10.0 mm] and positioned at the origin (0,0,0) of Blender’s world coordinate system. This scaling approach eliminated the need for individual camera adjustments for each object during rendering. The camera is then placed at a distance of 36.0 mm along the positive *Y* axis of the world coordinate system, as illustrated in Figure 7. Due to the limitations of top-down views in revealing the full extent of deformation for most of the studied objects, it can be difficult to visually distinguish the geometric variations. To address this, we selected unique camera positions for each deformed model during rendering. These positions were chosen to maximize the visibility of the deformation in the final images. The specific camera positions used for each object are detailed in Table 4.

Table 4 details:The rotation angle of the object around the *Z* axis of its coordinate system ($R_{Z_{O}}$), where *R* represents the rotation itself, *Z* indicates the axis of rotation, and *O* denotes the object being rotated.The position of the light along the *X* axis of the world coordinate system ($T_{X_{L}}$), where *T* represents the translation itself, *X* indicates the axis of translation, and *L* denotes the light being translated.The rotation angle of the camera around the *X* axis of the world coordinate system ($R_{X_{C}}$), where *R* represents the rotation itself, *X* indicates the axis of rotation, and *C* denotes the camera being rotated.

Each of the three parameters is defined as a series of numbers, using the format {start value: increment value: stop value}. By varying all possible combinations of these values, a unique series of images is generated for each object. The number of images produced for each object, using these parameters, is specified in the table.

Therefore, all position and rotation values of the objects, except $R_{Z_{O}}$, remain constant at zero during rendering. While $T_{X_{L}}$ governs the position of the light during rendering, other position and rotation parameters are held constant at zero, with the exception of the position along the *Z* axis of the world coordinate system, $T_{Z_{L}}$, which is fixed at 40.0 mm. The trajectory of the camera is defined as a curve on an imaginary hemisphere along the *Y* axis of the world coordinate system. The origin of the hemisphere coincides with the origin of the world coordinate system, and its radius, *R*, is set to 36.0 mm. To follow this trajectory, the $R_{X_{C}}$ parameter is utilized as the input for the following equations: (1)$$T_{Z_{C}}=R\;\cdot \;\sin\left(R_{X_{C}}\right),T_{Y_{C}}=R\;\cdot \;\cos\left(R_{X_{C}}\right)$$
where $T_{Z_{C}}$ and $T_{Y_{C}}$ represent the camera’s position along the world coordinate system’s *Z* and *Y* axes, respectively. During rendering, all other camera position and rotation parameters are fixed at zero. To improve the generalization capability of the training model, we strategically varied the camera and light positions during rendering. This process, as illustrated in Figure 8, generates images that closely resemble real-world photographs of the deformed objects. This is achieved by varying the illumination intensity and the background of the object during rendering. By training on these more realistic images, the model learns features that are transferable to unseen real-world data. In simpler terms, the model becomes adept at recognizing patterns in real images of deformed objects because it has been exposed to a wider variety of simulated views during training.

Therefore, $R_{Z_{O}}{(}^{\circ })$ is the only parameter of the object whose value changes during the rendering process, and other values are constant at zero during the process. Regarding the camera position and rotation, except for its position along the *Y* and *Z* axis and its rotation angle around the *X* axis of the world coordinate system, all other parameters of position and rotation are constant at zero during the rendering process.

Figure 8 illustrates the impact of varying camera and light positions on the rendered images. First row (Figure 8a–d): This row shows how the illumination intensity of the rendered images changes as the light position is shifted along the *X* axis of the world coordinate system from 0 mm to 45 mm, while the camera rotation around the *X* axis (denoted by $R_{X_{C}}$) is fixed at 60°. Second row (Figure 8e–h): This row demonstrates the effect of varying the camera rotation angle around the *X* axis (denoted by $R_{X_{C}}$) from 0° to 30°. Here, the light position along the *X* axis (denoted by $T_{X_{L}}$) is held constant at 0 mm. Consequently, the light intensity also remains constant. During each rendering step, the positions and orientations of both the rendered object and the camera are saved to a text file for later use during model training.

#### 3.1.3. Real-Data Generation

To assess the performance of both the proposed deformation estimation model and the label generation algorithm, we created a real-image dataset of actual deformed models. This limited dataset includes 3D-printed actual deformed models; and real-world images of the printed models. We leveraged a Prusa i3 MK3S+ 3D printer and polylactic acid (PLA) material to construct actual deformed objects. These objects were then captured from various viewpoints. The Prusa printer’s inherent precision of 0.15 mm aligns well with the objectives of this research. However, due to the printer’s printing time (approximately 15 h per object), we opted to create a dataset of 20 deformed objects: 10 skateboard models and 10 vent cover models. Figure 9 showcases four of these printed models as an example.

After printing the deformed objects, we devised a setup to capture their images as depicted in Figure 10. This setup utilized an Intel RealSense D435 camera mounted on a KUKA robot arm (KUKA KR 6 R700 sixx) for maneuverability. The deformed model was positioned directly in front of the camera, and 20 images were captured for each model from various viewpoints. In total, 200 images were captured for each of the bracket and vent cover. Our access to the 3D mesh models of the deformed objects provided a significant advantage. We could directly access the 3D location of each vertex and the exact dimensions of each model. This information, along with the manually extracted pixel locations from the captured images corresponding to these 3D points, allowed us to generate a set of 3D–2D point correspondences. These correspondences were then used to estimate the extrinsic camera parameters (three translation and three rotation parameters) relative to the object coordinate system using the EPnP method [30]. EPnP is a popular method for estimating a camera’s extrinsic parameters (translation and rotation) relative to the world coordinate system.

We employed the MATLAB camera calibrator toolbox [31] to determine the intrinsic parameters of the camera. This toolbox leverages the Zhang method [32] and utilizes captured images of a checkerboard pattern for calibration. The detailed camera calibration results are presented in Table 5. To achieve accurate calibration, we captured approximately 137 images of the checkerboard pattern from various viewpoints using the Intel RealSense camera. The camera was configured to capture images at a resolution of 1280×720 pixels, where 1280 represents the image width (*u*) and 720 represents the image height (*v*) in pixels.

Before feeding the captured images into the training model, we undistorted them to compensate for lens distortion. The distortion parameters were obtained during camera calibration using the MATLAB camera calibrator toolbox. These parameters include:Radial distortion coefficients: $\left[k_{1}=0.1731,k_{2}=-0.5222,k_{3}=0.4402\right]$;Tangential distortion coefficients: $\left[p_{1}=0.0012,p_{2}=-0.001\right]$.

Table 5 presents both the principal point and focal length values. These values are defined in pixels, with the first focal length value ($f_{u}=907.7907$) corresponding to the focal length in the *U* (horizontal) direction and the second value ($f_{v}=907.9046$) corresponding to the focal length in the *V* (vertical) direction of the pixel coordinate system of the captured image. The origin of this coordinate system is located in the upper-left corner of the image where the *U* direction is towards the width and *V* towards the height of the image. Similarly, the principal point values specify the image location where the optical axis intersects the image plane. The first value ($u_{0}=630.1808$) refers to the horizontal coordinate, and the second value refers to the vertical coordinate ($v_{0}=371.5688$). The camera calibration process achieved a mean reprojection error of 0.1471 pixels. This low error indicates high accuracy and suggests that the estimated calibration parameters are reliable and precise.

The final captured real images using the Intel RealSense, which are undistorted and cropped to a resolution of 600×600 to meet the training model input requirements, are illustrated in Figure 11. Unlike rendered images, which are rendered from predetermined points of view, real images are captured from arbitrary points of view just by prioritizing the visibility of the deformation.

### 3.2. Label Generation

In machine learning, labels provide crucial ground truth information that guides a model’s training process. The model’s loss function minimizes the difference between its predictions and these labels, effectively adjusting its internal parameters (weights) to complete the training process. The initial labels generated by the Ansys software are incompatible with our training model. This is because Ansys automatically creates a mesh for static force analysis with a different (higher number and positioning) vertex configuration compared to the initial mesh input of the model (Figure 12). This presents a challenge for training graph convolutional networks (GCNs), which excel at learning from graph-structured data but are sensitive to the number of vertices in the input graph from a computational standpoint. The high vertex count in Ansys meshes significantly increases training costs. Therefore, this work proposes a label generation process to control the number of vertices while preserving essential information for the GCN model.

As shown in Figure 12a, the deformed vent cover mesh exhibits a significantly higher vertex count (6268) compared to the undeformed initial model (Figure 12b), with a nearly six-fold increase. This observation is consistent across all studied objects. Consequently, directly using the high-vertex-count Ansys output meshes as input for GCN models becomes computationally prohibitive. Furthermore, a higher number and different positioning of vertices poses an additional challenge for employing the prevalent chamfer distance (CD) algorithm [33] for error calculation between two 3D point clouds as formulated in the following equation: (2)$$CD\left(A,B\right)=\frac{1}{\left|A\right|}\sum_{a\in A}\min_{b\in B}{∥a-b∥}_{2}^{2}+\frac{1}{\left|B\right|}\sum_{b\in B}\min_{a\in A}{∥b-a∥}_{2}^{2}$$
where *A* and *B* represent two point clouds containing points denoted as *a* and *b*, respectively. The chamfer distance relies on a nearest-neighbor search, which can lead to erroneous results for highly deformed meshes; it iterates through each point in one point cloud, searching for the nearest point in the other point cloud. The closest point is then assigned as the corresponding point to the selected point, and the distance between them constitutes the chamfer distance for that selected point. This process continues until all points in both point clouds have been considered. The challenge arise from the fact that for each vertex in the undeformed mesh, there might be multiple candidate vertices in the deformed mesh with similar distances. This ambiguity can cause the nearest-neighbor search algorithm to converge on a local minimum, mistakenly assigning an incorrect corresponding vertex from the deformed mesh to the undeformed one. This error becomes more pronounced with significant deformations, as Figure 13 illustrates.

In Figure 13, two distinct 2D point distributions are depicted. The true corresponding points of the blue distribution are connected with green dashed lines for clarity. The presumed deformed distribution of 2D points consists of eleven points highlighted in magenta, conforming to a Gaussian distribution. Concurrently, the initial distribution is represented by three blue points arranged along a straight horizontal line. It is discernible that the magenta points, symmetrically positioned on either side of the curve, exhibit relatively minor deformation compared to those situated at the apex of the curve concerning the initial distribution.

While a nearest-neighbor search strategy effectively identifies corresponding points for instances located on either side of a linear distribution (blue points), it can lead to inaccurate matches for central points (red dashed lines) due to inherent limitations. This discrepancy arises because varying degrees of deformation along the curve impact the effectiveness of nearest-neighbor search in establishing accurate correspondences. This limitation generalizes to 3D point clouds as well.

Hence, the chamfer distance algorithm excels in identifying optimal corresponding points when deformations are marginal. However, direct application of the CD to meshes with significant deformations may not result in accurate corresponding points. This rationale explains the deformation step values selected in Table 2 for the different studied objects, ensuring that each deformed model exhibits marginal variation in geometry relative to the previous deformed model. Our experience indicates that values below 0.5 mm provide sufficient geometric variations for the proposed label generation algorithm. In response to this challenge, we propose a sequential deformation (SD) approach to generate the optimal mesh for each deformed object. The comparative outcomes of directly applying the chamfer distance algorithm to deformed mesh models with significant deformations, as opposed to employing our proposed sequential strategy on the same deformed models, are presented in Figure 14.

As illustrated in Figure 14f, directly applying the chamfer distance algorithm to the initial undeformed model and the most deformed model results in significant labeling errors. These errors include:Loss of circularity in the central features;Flipped faces leading to incorrect surface shading;Near-zero thickness in some regions;Generally bumpy surface texture.

In contrast, the proposed sequential deformation method generates a high-quality label (Figure 14g) that accurately capture the properties of the deformed object. This label preserves the smoothness of the surface, the circularity of the central features, the object’s thickness, and the correct orientation of faces.

Our sequential deformation method demonstrates the ability to generate high-quality meshes suitable for various applications, like object quality inspection and 3D reconstruction-based object detection. By utilizing the proposed label generation method, the achievement of high-quality objects becomes feasible, while the computational cost of training is reduced by eliminating the computationally expensive CD algorithm from the training loop. Figure 15, Figure 16, Figure 17, Figure 18 and Figure 19 compare the results obtained using the SD method and the CD algorithm. The first column from the left presents the undeformed CAD models for reference. The second column shows the deformed objects generated by the Ansys software, which have a high number of vertices and faces. Both CD (third column) and SD (fourth column) algorithms aim to represent the high-vertex, high-face, deformed models using the mesh structures (vertex count and face order) of the initial, undeformed objects. This comparison allows for evaluation of how effectively each algorithm captures the true deformations.

As shown in Figure 15, Figure 16, Figure 17, Figure 18 and Figure 19, directly applying the chamfer distance algorithm to highly deformed models leads to significant errors. These errors include flipped faces, near-zero thickness in some regions, and loss of circularity in circular features, all of which are evident across various studied objects with high geometric variations. This demonstrates the limitations of the CD algorithm for precise and reliable correspondence finding, rendering it unsuitable for the loss function of a 3D point cloud location prediction model.

To address these limitations, this paper proposes the sequential deformation method. This method accurately identifies corresponding points between the undeformed and deformed models’ vertices. Consequently, it generates a precise representation of the deformed model by leveraging the undeformed mesh structure. By employing the sequential deformation method for labeling deformed objects, we specify a target 3D location for each vertex of the initial undeformed model. This eliminates the need for the chamfer distance algorithm within the training model’s loss function. By identifying the corresponding points beforehand using sequential deformation, we avoid the potential for the model to become stuck in local minima during training, which could lead to inaccurate mesh generation. This approach also reduces the computational cost associated with the training process and guarantees convergence to the optimal solution.

### 3.3. Model Design

In this paper, mesh models are represented as a graph G=(V,E) to leverage graph convolution for extracting features from a graph, as introduced in [34]. In this formulation, $V=\{v_{1},v_{2},\ldots ,v_{n}\}$ and $E=\{\left(v_{i},v_{j}\right)\;|\;v_{i},v_{j}\in V\}$ denote sets of *n* vertices and edges, respectively. Each vertex can be associated with a feature vector, and using the function $f:V\to R^{m}$ these features are mapped to each vertex. Here, $R^{m}$ represents the m-dimensional Euclidean space, with *m* denoting the number of features in each vector. The graph convolution operation is then expressed as
(3)$$f{\left(v_{i}\right)}^{\left(l+1\right)}=\sigma \left(\sum_{v_{j}\in N\left(v_{i}\right)}W^{\left(l\right)}\;\cdot \;f{\left(v_{j}\right)}^{\left(l\right)}+bias^{\left(l\right)}\right)$$
where $f{\left(v_{i}\right)}^{\left(l+1\right)}$ and $f{\left(v_{j}\right)}^{\left(l\right)}$ represent the feature vectors of vertices $\left(v_{i}\right)$ and $\left(v_{j}\right)$ for layers *l* and l+1, respectively. $W^{\left(l\right)}$ and $bias^{\left(l\right)}$ are the weight matrix and bias vector for layer *l*, respectively. Finally, σ is the activation function, and $N(v_{i})$ denotes the neighbors of vertex $v_{i}$ in the graph.

Inspired by the Pixel2Mesh model [3], our training model incorporates a novel design to estimate object deformation from a single image and a loss function specifically designed for aligning with the generated labels. Given an initial mesh model of the object, an RGB image of the deformed object, and the camera’s intrinsic and extrinsic parameters, this end-to-end learning method predicts vertex offsets or deformation as output.

The model consists of three key components: 2D and 3D feature extractors (CNN and GCN blocks, respectively) and projection layers, as depicted in Figure 20. The pipeline begins by extracting 2D feature maps from a distortion-free image using a VGG-16-like architecture [35]. This architecture consists of four convolutional blocks, each containing several 2D convolutional layers with ReLU activations (Figure 20, CNN block, blue-colored cubes). The first two blocks have dimensions of (600,600,64) and (300,300,128), respectively. The final two blocks have three layers with dimensions of (150,150,256) and (75,75,512), respectively. The CNN architecture incorporates three strategically placed max-pooling layers with a pool size and stride of (2,2) (Figure 20, CNN block, yellow-colored cubes). These layers follow each convolutional block except the last, effectively reducing the spatial dimensions (height and width) of the feature maps. This design aims to capture a wide range of features, from low-level geometric details to high-level semantic information.

The four resulting feature maps are then fed into the projection layer to register the corresponding 2D features with each 3D vertex. The projection layer is responsible for projecting the initial mesh’s vertices into the feature maps extracted from the CNN block. To achieve precise vertex projection, the layer receives the initial mesh vertex locations and the camera parameters. These camera parameters include both extrinsic (object’s translation and rotation relative to the camera) and intrinsic parameters (focal length and principal point of the camera used for image rendering). It is important to note that since the rendered images are distortion-free, the intrinsic parameters do not account for distortion.

A key challenge lies in balancing computational efficiency with ensuring unique feature representation for each vertex. While each vertex has a unique 3D location and ideally requires a large feature map for registration, using very high resolution images becomes computationally expensive. Due to the very low resolution of input images, vertices are often registered with identical features, as there are not enough distinct 2D features available. To address this, the model utilizes a 600×600 image size, striking a balance between computational cost and capturing valuable information for each vertex through the 2D convolution process within the CNN block.

The registered corresponding features from the four feature maps are concatenated, along with the initial 3D features of vertices (representing the 3D location of each vertex), to create a new feature vector for each vertex. The final shape of the feature matrix is denoted as n×d, where *n* represents the number of vertices in the initial mesh, and d=963 is the channel of the vertex features. This channel is the sum of the feature channels from each of the four 2D feature maps, totaling 960 channels, along with the channel of the initial 3D features, which is 3.

The projection layer feeds its output, a feature matrix, into a graph convolutional network (GCN) block (Figure 20, GCN block). This GCN block aims to extract 3D features from the input mesh, considering its structure as a graph. It consists of seven graph convolutional layers with an ReLU activation function (Figure 20, green rectangles), followed by four dense layers (Figure 20, magenta rectangles). The first three graph convolutional layers have dimensions of (n,512), and the remaining four layers have dimensions of (n,256), where *n* represents the initial mesh vertex count. While graph convolutional layers are responsible for extracting 3D features from a graph by considering the graph structure, fully connected (dense) layers focus on making high-level predictions for each vertex offset based on the input 3D features of a vertex. The subsequent four dense layers have progressively decreasing dimensions: (n,128),(n,64), and (n,32), all with ReLU activation. The final dense layer has a linear activation function and a dimension of (n,3), where 3 represents the offset values for a vertex’s location in 3D space. These predicted offsets are then used to update the initial 3D positions of the mesh vertices.

Given that the input image corresponds to the deformed model of the input mesh, the accuracy of the projection process becomes compromised. This is because the 3D locations of the initial mesh vertices differ from those of the deformed model due to the applied deformations. To mitigate this issue, we propose a model that incorporates two projection and GCN blocks. The first GCN block predicts offsets to update the 3D locations of the initial model’s vertices. These updated vertices now have more accurate positions relative to the deformed model represented in the input image. Subsequently, the second projection layer utilizes these refined vertices. By employing this double prediction and projection approach, we enhance the training model’s prediction accuracy, particularly by improving the precision of the feature registration process. Using the final prediction of the model, the locations of the vertices are updated with the predicted offsets. To reconstruct the mesh of the updated vertices, the face information of the initial mesh model is then leveraged.

Since the training model operates as a regression model, the Euclidean distance is used to compute the error between the predicted and labeled point clouds. The predicted point cloud is obtained by adding the predicted offsets to the initial mesh vertices’ locations. Both the prediction and the label consist of point clouds with an equal number of points, each represented by its 3D location in a matrix format with dimensions n×3, where *n* corresponds to the number of vertices matching the initial mesh. Given the prediction and label vertices as $P=\{p_{1},p_{2},\ldots ,p_{n}\}$ and $Q=\{q_{1},q_{2},\ldots ,q_{n}\}$, respectively, the loss of the model is formulated as the following equation:(4)$$Loss=\frac{1}{n}\sum_{i=1}^{n}∥p_{i}-q_{i}∥$$
which calculates the mean value of the Euclidean distance between two 3D point clouds. With the prepared image dataset of deformed objects, our proposed sequential deformation method for generating precise labels for each deformed model, and a trained model capable of learning the connection between input RGB images and mesh vertex offsets, we possess all the necessary tools to test our methodology. Details of this implementation and its results are presented in the following section (Section 4).

## 4. Results

This section is dedicated to detailing the setup employed for training the model, along with the evaluation process conducted on the generated dataset.

The model is implemented using the TensorFlow and Keras frameworks on the Google Colaboratory (Colab) platform, with the generated dataset uploaded to Google Drive. The Drive is then mounted to Google Colaboratory to facilitate data access. Since the number of vertices varies among the studied objects, the training process requires adjustments for each object. While the core model architecture remains constant, the model’s inputs, batch size, and data size are different for each object’s training model. Consequently, training and testing times (both for the entire test dataset and per instance), along with the number of training epochs, exhibit corresponding variations, as detailed in Table 6.

Training and testing times are primarily influenced by object complexity (reflected by the number of vertices), training/testing data size, and available computational resources. This study utilizes the CPU and RAM provided by Colab. While the model can run on GPUs, limited RAM on Colab’s GPUs restricts batch size, making CPU execution preferable. Equipping the model with powerful GPUs and ample RAM has the potential to accelerate training times significantly. As shown in Table 6, the RFRL model has the shortest training time due to its small dataset. Conversely, the vent cover model has the longest training time due to its complexity (reflected by the highest vertex count of 1145) and large dataset. Interestingly, despite having more data, the skateboard model trains faster than the vent cover due to its lower vertex count (443). Table 6 reports the model’s processing times in two ways: total execution time for the entire test dataset (reported in hours) and processing time per object within the test dataset (reported in seconds). The key takeaway is the model’s efficiency in evaluating individual objects. As shown in the results, the average processing time for each object in the test dataset is approximately 1.5 s.

During the model training process, the hyperparameters are carefully tuned with the following configurations: the optimization algorithm is Adam, the initial learning rate is set to $2\times 10^{-4}$, the learning rate decreases every 20 epochs, and the decay rate is 0.2. For numerical stability, both model predictions and labels are normalized to the range [−1.0,1.0 mm]. To balance the impact of the Euclidean loss with regularization and mitigate the influence of very small errors, the loss is scaled up by a factor of 4000 for all trained models.

We assess model performance using the mean (denoted by $M\;Acc^{★}$) and standard deviation (denoted by $Sdev\;Acc^{★}$) of the Euclidean distance between predicted vertex offsets and corresponding ground truth meshes for all models in the test set. Both predicted offsets and ground truth labels are normalized to the range [−1.0,1.0 mm]. This normalization ensures the mean Euclidean distance also falls within this range, with a value of 1.0 mm representing the maximum error and 0 indicating a perfect match.

A key benefit of normalized Euclidean distance is its scale invariance. This allows for comparisons between models applied to objects of different sizes within the study. To obtain the error in real-world measurements, we multiply the normalized values by a scale factor specific to each object. These scale factors represent half of the assumed maximum dimension (e.g., diameter for a vent cover) of a manufactured object. For example, the skateboard component has a scale factor of 132.264, while the vent cover’s factor is 122.237. The bracket, oval flange, and RFRL objects have scale factors of 100.0, 160.0, and 125.02, respectively. By multiplying the normalized accuracy by the corresponding scale factor, we obtain the scaled error in millimeters, providing a more intuitive understanding of prediction accuracy for real-world objects.

These scale factors are determined by assuming specific values for the largest dimension of each object if it were to be manufactured. In this study, the scale factor corresponds to half the assumed maximum dimension since the prediction is normalized to the range [−1.0,1.0 mm]. For instance, the vent cover’s scale factor of 122.237 translates to a presumed diameter of 244.474 mm for a manufactured vent cover. This method can be applied to calculate the actual largest dimension of other studied objects.

Table 7 summarizes the normalized mean and standard deviation of the Euclidean error for all studied objects. The trained model achieved its best performance in estimating the deformation of the RFRL object, with a normalized mean Euclidean error of $M\;Acc^{★}$ = 0.000699 mm. However, its accuracy decreased to $M\;Acc^{★}$ = 0.002403 mm for the oval flange object.

To assess the trained model’s performance, we evaluated it on a set of deformed models for each object studied in this paper. The results are detailed in separate tables: Table 8 for the skateboard, Table 9 for the bracket, Table 10 for the RFRL, Table 11 for the oval flange, and Table 12 for the vent cover. In all tables, the “applied force” parameter represents the magnitude of a force applied along the object’s *Z* axis (in its world coordinate system) during a static force simulation using the Ansys software. The “MD” parameter indicates the magnitude of the maximum deformation caused by the applied force. As previously mentioned, the mean and standard deviation of the normalized Euclidean error are summarized under the parameters $M\;Acc^{★}$ and $Sdev\;Acc^{★}$, respectively. Their corresponding real-world values are denoted by MAcc and SdevAcc, respectively.

For training the neural network levering the skateboard data, 22 deformed models of the skateboard were selected, with a maximum deformation range of MD=[−19.93 mm,19.93 mm] and a step change of 1 mm (Table 8). Notably, the model’s average accuracy in predicting the actual object’s deformation remained consistently below 0.3 mm for all trained models. This performance is particularly impressive considering the maximum dimension of a real-world skateboard is assumed to be approximately 264.528 mm, and sub-millimeter accuracy is considered acceptable in most manufacturing standards. Figure 21 and Figure 22 present two key inputs of the training model: the undeformed mesh and the image of the deformed component, along with the corresponding predicted mesh (the output of the training model). Specifically, Figure 21 pertains to a skateboard subjected to F=33.5 N and MD=15.0 mm, while Figure 22 pertains to a skateboard subjected to F=−33.5 N and MD=−15.0 mm. The outputs are depicted in terms of the actual Euclidean error for each vertex, which quantifies the error of each vertex in the predicted mesh relative to the corresponding vertices in the ground truth mesh.

A dataset of 20 deformed bracket objects was selected to train the training model. As is evident from Table 9, the magnitude of variation in the deformed objects’ geometry differs in the range of MD=[−14.15 mm,14.15 mm], with a step size of approximately 1.0 mm. The mean accuracy (Euclidean error) of the actual model, with the largest dimension (length of the bracket) of 200.0 mm, is consistently lower than 0.2 mm for all the trained deformed models. Given the length of the object (200.0 mm), this sub-millimeter result is notably accurate.

Figure 23 and Figure 24 illustrate the inputs and outputs of the training model for two instances of the bracket with the following specifications: first instance: [F=28 N,MD = 6.00 mm]; second instance: [F=−46 N,MD=−9.86 mm]. The inputs to the training network consist of the initial undeformed mesh of the bracket and an image of the deformed bracket. The predicted mesh is displayed from different viewpoints, with a color map indicating the error magnitude of each vertex in the prediction model.

Table 10 presents the results of training the network on data from the RFRL object, which include 20 deformed components with maximum deformations ranging from [6.09 mm,24.97 mm], in approximately 1.0 mm increments. The training model’s accuracy on this dataset is notable, with the average Euclidean error on the real-world object, with a diameter of 250.041 mm (largest dimension), being consistently below 0.1 mm for all the deformed models in the RFRL dataset.

In Figure 25 and Figure 26, we present the predicted meshes of the deformed RFRL object from various viewpoints. These rendered images use colors to indicate the Euclidean error magnitude between the predicted vertex positions and their corresponding ground truth locations in the labeled mesh for each vertex of the predicted model. The specifications for the two instances of the RFRL object depicted are as follows: the first instance has a force of F=2750 N and maximum deformation of MD=15.96 mm, while the second instance has a force of F=3950 N and maximum deformation of MD=15.96 mm. To enhance understanding of the training model’s input data, the figures also display an initial undeformed version of the RFRL object alongside an image of its deformed version.

Among the deformed models of the oval flange object, 20 parts with varying magnitudes of geometric variation were selected to train the designed model. These variations range from −18.20 mm to 18.20 mm, with a step size of 1.0 mm. As shown in Table 11, the actual mean Euclidean error is below 0.4 mm for most of the trained models. Given the length (largest dimension) of the actual oval flange, which is 320.0 mm, the prediction accuracy is acceptable, as it falls within the sub-millimeter threshold, which is notable for an object of this size.

Two samples of the oval flange predictions are illustrated in Figure 27 and Figure 28. The first sample, shown in Figure 27, has an applied force of F=1750 N and maximum deformation of MD=12.02 mm. The second sample, shown in Figure 28, has an applied force of F=−2200 N and maximum deformation of MD=−15.11 mm. In these figures, the vertices of the predicted meshes are color-coded to represent the Euclidean error between the predicted vertex locations and their corresponding locations in the ground truth meshes. Additionally, the figures include the input mesh model used for training the network, as well as an image of the deformed oval flange, providing comprehensive details of the training model’s inputs.

To train the vent cover object using the proposed model, 22 deformed models were selected, with maximum deformations ranging from −13.97 mm to 13.97 mm, in increments of 1.0 mm. The results in Table 12, regarding average Euclidean error, show that the model’s accuracy for the vent cover object was below 0.4 mm for most of the examined models. Given that the diameter (largest dimension) of the actual object is 244.375 mm, this prediction accuracy is acceptable, as it meets the sub-millimeter criterion.

Figure 29 and Figure 30 display the accuracy of the training model’s predictions for each vertex of the two selected parts using a color gradient. The first part has an applied force of F=310 N and maximum deformation of MD=8.02 mm, while the second part experiences a force of F=−390 N and maximum deformation of MD=−10.09 mm. The figures also include the initial undeformed mesh model and an image of the deformed model, which serve as inputs for the training network.

As our model is a regression model predicting continuous values, evaluating its accuracy on the training dataset for all studied objects requires defining a threshold. To assess the model’s ability to capture deformations effectively, two thresholds based on the actual Euclidean distance between predicted and ground truth meshes were employed: 0.4 mm and 0.2 mm. These thresholds were chosen to prioritize sub-millimeter accuracy while also providing insights into the model’s overall performance. At the first threshold (0.4 mm), the model achieved a high precision of 0.96 across all studied objects. This indicates that for a large portion of the objects, the model’s predicted deformations were within 0.4 mm of the actual ground truth values. However, as the threshold was tightened to 0.2 mm, the precision naturally decreased to 0.53. This suggests that while the model can achieve sub-millimeter accuracy in many cases, maintaining this level of precision for all objects is more challenging.

Generalization, a crucial aspect of any training model, allows it to perform effectively on unseen data. Striking a balance between achieving good performance on unseen data and the training set can be challenging. In this work, we prioritized generalizability starting from the dataset design and continuing through model training.

During the creation of the deformed object dataset, we incorporated variations in background and illumination intensity for the rendered images. This approach aims to make the images resemble real-world scenarios, enhancing the model’s ability to adapt to unseen data.Regularization and early stopping techniques were employed during model design to improve the model’s generalizability. Regularization helps prevent overfitting by penalizing overly complex models, while early stopping halts training before the model memorizes the training data excessively. By leveraging a validation dataset, we continuously monitored how well the model performed on both the training and unseen validation data. This allowed us to identify the optimal training point and prevent overfitting on the training data.

To assess the model’s generalization capability to unseen real-world data, we prepared a small dataset of real images of 3D-printed, deformed objects, as detailed in the real-data generation section (Section 3.1.3). Since 3D printing deformed objects can be time consuming, and our primary goal here is to evaluate generalization, this small dataset consisting of the vent cover and bracket was sufficient for our purposes. Each object in the real dataset has 20 images captured from various arbitrary viewpoints. We leverage 15 of these images for training the network and reserve the remaining five images for testing. To improve the model’s performance on real-world data, we fine-tuned the parameters obtained by training on synthetic data using a combined dataset containing both synthetic and real images. Table 13 summarizes the details of the model’s evaluation on the test set of real data for the bracket objects, while Table 14 presents the results for the vent cover’s real data.

Figure 31 and Figure 32 visualize the Euclidean error between the predicted mesh vertices and their corresponding ground truth labels using a color map. The predicted mesh is rendered in color from different viewpoints to enhance the visualization of the error distribution. For reference, these figures also depict the input mesh and the real image of the 3D-printed deformed objects used as input to the training model.

It is evident from Table 13 and Table 14 that the average Euclidean error of the real bracket and vent cover components is approximately between 1.0 and 2.5 mm. This result is acceptable considering the low amount of real data (only 15 images) as well as the length of the actual bracket (200 mm), and the diameter of the 3D-printed vent cover (244.375 mm). Moreover, capturing real-world images introduces additional error sources compared to synthetic data. These errors include camera calibration inaccuracies, camera pose estimation errors, and image distortions. Unlike synthetic images, which are free from these issues, real-world image errors can contribute to the deformation prediction errors in the training model. This happens because such errors degrade the accuracy of the features extracted from the images. Noteworthy is that the results evaluate the generalization capability of the network on real data and, given the amount of real data, the results are reasonable. This accuracy of 3D information of the predicted meshes may even meet the requirements of some manufacturers for their inspection process.

This section comprehensively analyzes the performance of the training model on test data for various objects, presenting detailed results in tables and figures. It is important to note that the color-coded images of predicted meshes (e.g., Figure 23e and Figure 25e) may exhibit non-smooth color transitions. This is because the colors represent the Euclidean error magnitude for each vertex relative to the corresponding ground truth mesh vertex, and neighboring vertices might have different errors, leading to distinct colors. Although the shading algorithm used by the Blender software for rendering these meshes interpolates between vertex colors within a single face to assign a color to that face and provide a smooth color transition, the distinct color of neighboring vertices can lead to discontinuities in the final rendered images.

## 5. Discussion

The proposed training model achieved sub-millimeter accuracy in estimating the actual deformations of all five deformed objects within the dataset. This success can be primarily attributed to the introduced sequential deformation method and design of the model. This method generates highly precise mesh labels for deformed objects while preserving the initial undeformed mesh’s vertices and face structure. As Figure 15, Figure 16, Figure 17, Figure 18 and Figure 19 demonstrate, the sequential deformation method exhibits exceptional accuracy and quality, particularly when dealing with highly deformed meshes. It consistently outperforms the commonly used chamfer distance algorithm. The method excels at preserving even fine geometric details of objects, including small features like holes. This is because the sequential deformation algorithm establishes highly accurate correspondences between the initial mesh and the deformed mesh, even when the magnitude and order of vertices differ significantly. As a result, the generated mesh labels possess smooth surfaces, facilitating the model’s accurate estimation of deformations.

Thanks to the two-step projection and dense layers, the designed training model achieved an actual accuracy below MAcc=0.4 mm in most of the examined deformed models (Table 8, Table 9, Table 10, Table 11 and Table 12). This accuracy provides precise 3D information regarding the geometry of the deformed objects, which can be utilized in any computer vision tasks requiring this information. In less geometrically complex components like the skateboard, bracket, and RFRL, where the geometry does not include significantly large holes, the results were exceptionally outstanding with the average actual Euclidean error below MAcc=0.2 mm in most instances. Utilizing the SD method allowed us to use initial meshes with a low number of vertices and generate mesh labels during a preprocessing stage. Consequently, this approach excluded the computationally expensive CD method from the loss function of the training model. This measure significantly reduced the average execution time to approximately 1.5 s per instance of test data for all studied objects. The model generalized well on the real data, where the actual Euclidean error was below MAcc=2.0 mm in most instances (Table 13 and Table 14).

Given that each deformed model only has 15 real images to fine-tune the pretrained model, and considering the extra sources of errors such as camera calibration and camera pose estimation process errors introduced in real data, this result is significant. It can be leveraged in applications that do not have stringent criteria for 3D information accuracy. Since collecting real data is a time-consuming process, applications that find an accuracy below MAcc=2.0 mm acceptable can leverage our model. They can train it on synthetic data, and then, fine-tune it using only a few real images, which can save a substantial amount of time.

A key strength of our approach lies in its material, deformation, and geometry independence. Both the sequential deformation algorithm and the training model rely solely on 3D vertex locations. This allows them to function effectively regardless of the material, type of deformation (plastic or elastic), or specific object geometry, as long as the deformations are visually discernible in the image. Our study demonstrates the model’s ability to handle deformations of varying magnitudes and directions in five objects with diverse shapes and geometries. This suggests broader applicability to a wider range of objects, provided the captured images sufficiently capture the introduced geometric variations.

This work focused on developing a model that relies solely on a single RGB image for deformation estimation. While the generated dataset can provide multiple images of a deformed model from various viewpoints, leveraging this additional information presents a promising avenue for future exploration. Using multiple views can potentially lead to more accurate and robust deformation estimation compared to single-image approaches. Furthermore, the model’s current scope is limited to RGB images. Industrial object inspection often utilizes other image modalities like thermal and monochrome imaging. Extending our approach to incorporate these image types has the potential to broaden the applicability of our work in real-world scenarios.

In this research, we assumed the object’s pose relative to the camera was known and provided within both rendered and real images. However, achieving pose invariance would significantly enhance the model’s practicality. Future work should investigate how the model can handle inaccurate initial pose information and refine its accuracy during the deformation estimation process. This could involve incorporating a loss function that simultaneously optimizes both pose and deformation errors.

Due to the time-intensive nature of dataset generation, this study evaluates the proposed methodology using only five objects with diverse geometries and deformations. Future research efforts could expand the dataset by introducing new objects with varying geometries, materials, deformation types, magnitudes, and directions. This expansion would enable a more comprehensive evaluation and validation of the methodology.

## Figures and Tables

**Figure 1 sensors-24-04707-f001:**
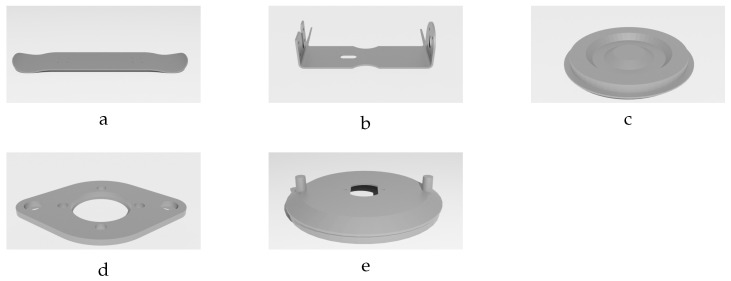
Images of the five textureless plastic objects studied in this paper: (**a**) skateboard, (**b**) bracket, (**c**) round flat receptacle lid (rfrl), (**d**) oval flange, and (**e**) vent cover.

**Figure 2 sensors-24-04707-f002:**
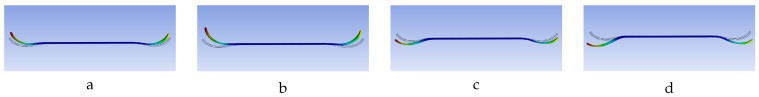
Images of the deformed versions of the skateboard; the wireframe illustrates the undeformed model: (**a**) F=22.5 N, MD=10.08 mm; (**b**) F=44.5 N, MD=19.93 mm; (**c**) F=−22.5 N, MD=−10.08 mm; (**d**) F=−44.5 N, MD=−19.93 mm.

**Figure 3 sensors-24-04707-f003:**
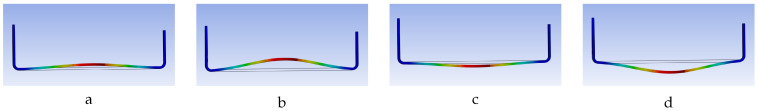
Images of the deformed versions of the bracket; the wireframe illustrates the undeformed model: (**a**) F=24 N, MD=5.14 mm; (**b**) F=66 N, MD=14.15 mm; (**c**) F=−24 N, MD=−5.14 mm; (**d**) F=−66 N, MD=−14.15 mm.

**Figure 4 sensors-24-04707-f004:**
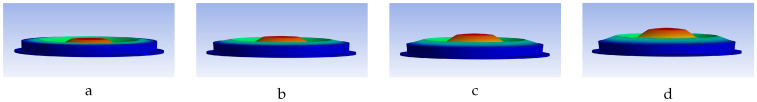
Images of the deformed versions of the round flat receptacle lid; the wireframe illustrates the undeformed model: (**a**) F=1050 N,MD=6.09 mm; (**b**) F=2250 N,MD=13.06 mm; (**c**) F=3450 N, MD=20.03 mm; (**d**) F=4300 N,MD=4300 mm.

**Figure 5 sensors-24-04707-f005:**
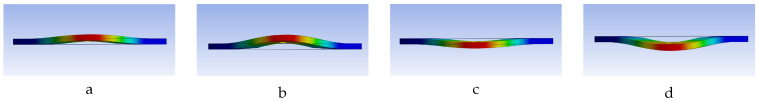
Images of the deformed versions of the oval flange; the wireframe illustrates the undeformed model: (**a**) F=1300 N, MD=8.93 mm; (**b**) F=2650 N, MD=18.20 mm; (**c**) F=−1300 N, MD=−8.93 mm; (**d**) F=−2650 N,MD=−18.20 mm.

**Figure 6 sensors-24-04707-f006:**
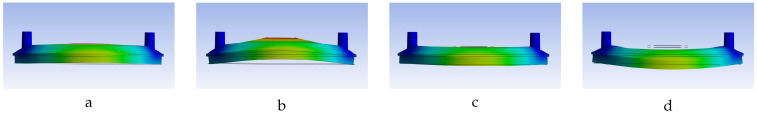
Images of the deformed versions of the vent cover; the wireframe illustrates the undeformed model: (**a**) F=150 N,MD=3.88 mm; (**b**) F=540 N, MD=13.97 mm; (**c**) F=−150 N, MD=−3.88 mm; (**d**) F=−540 N, MD=−13.97 mm.

**Figure 7 sensors-24-04707-f007:**
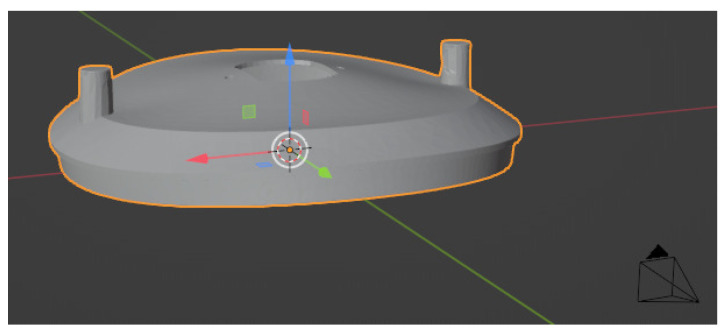
This schematic depicts the camera position relative to the deformed vent cover, located at the origin (0,0,0) of Blender’s global coordinate system. The red, green, and blue axes represent the *X*, *Y*, and *Z* axes, respectively.

**Figure 8 sensors-24-04707-f008:**
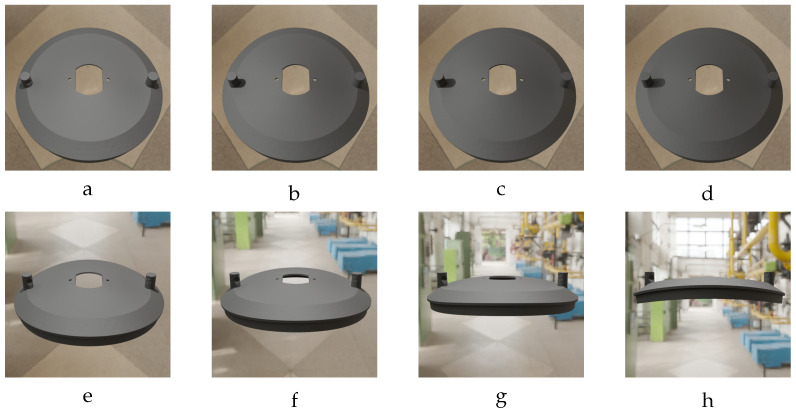
Images of the deformed vent cover with F=540.0 N, MD=13.97 mm, and $R_{Z_{O}}=0{}^{\circ}$. (**a**) $T_{X_{L}}=0\;mm$, (**b**) $T_{X_{L}}=25\;mm$, (**c**) $T_{X_{L}}=35\;mm$, (**d**) $T_{X_{L}}=45\;mm$, (**e**) $R_{X_{C}}=0{}^{\circ}$, (**f**) $R_{X_{C}}=10{}^{\circ}$, (**g**) $R_{X_{C}}=20{}^{\circ}$, and (**h**) $R_{X_{C}}=30{}^{\circ}$. If not mentioned, $R_{X_{C}}=60{}^{\circ}$ and $T_{X_{L}}=0\;mm$.

**Figure 9 sensors-24-04707-f009:**
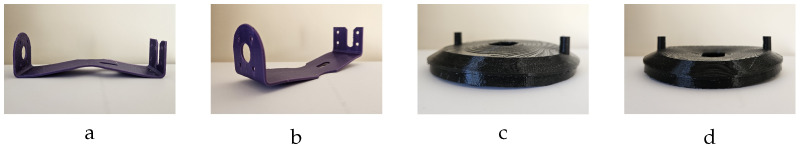
Images of the 3D-printed deformed models captured by a smartphone camera. (**a**) Deformed model of the bracket with F=66.0 N, MD=14.15 mm. (**b**) Deformed bracket component with F=−66.0 N, MD=14.15 mm. (**c**) Deformed vent cover component with F=270 N, MD=6.98 mm. (**d**) Deformed model of the vent cover object with F=−500.0 N, MD=−12.93 mm.

**Figure 10 sensors-24-04707-f010:**
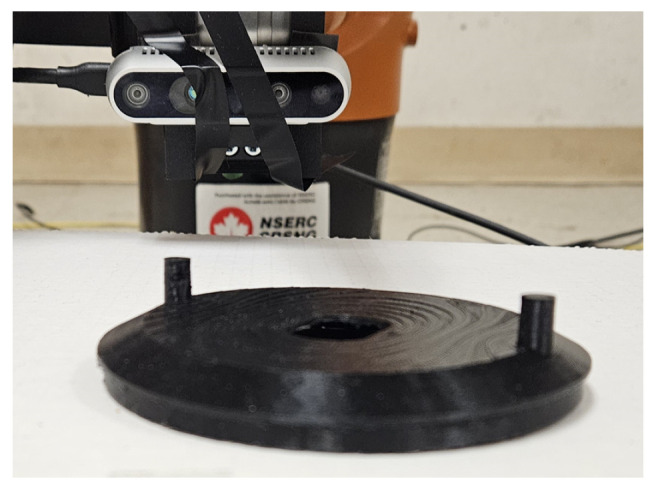
Setup devised to capture pictures from the printed deformed models.

**Figure 11 sensors-24-04707-f011:**
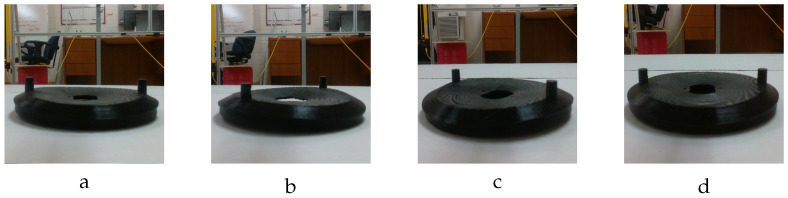
Real-world images of a 3D-printed, deformed vent cover (F=−500.0 N, MD=−12.93 mm) used for training the machine learning model. (**a**) $R_{Z_{O}}=-0.70{}^{\circ}$ and $R_{X_{C}}=85.36{}^{\circ}$. (**b**) $R_{Z_{O}}=36.46{}^{\circ}$ and $R_{X_{C}}=82.09{}^{\circ}$. (**c**) $R_{Z_{O}}=-34.40{}^{\circ}$, and $R_{X_{C}}=72.71{}^{\circ}$. (**d**) $R_{Z_{O}}=-3.49{}^{\circ}$ and $R_{X_{C}}=76.82{}^{\circ}$.

**Figure 12 sensors-24-04707-f012:**
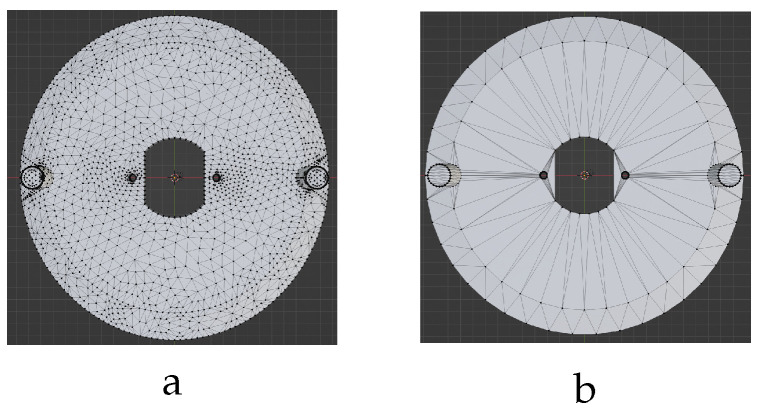
Mesh comparison: (**a**) Deformed mesh model, Ansys output with 6268 vertices and 12,544 faces. (**b**) Initial undeformed mesh model (training model input) with 1145 vertices and 2298 faces.

**Figure 13 sensors-24-04707-f013:**
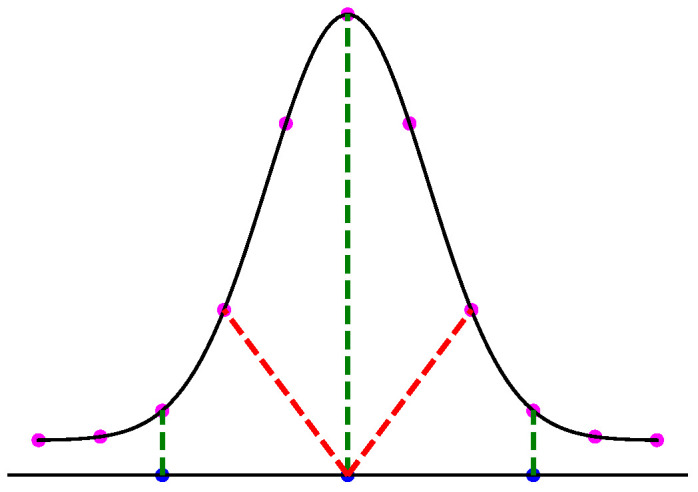
The chamfer distance algorithm applied on two 2D distributions. False correspondences (red dashed line) for middle blue point, and true correspondence (green dashed line).

**Figure 14 sensors-24-04707-f014:**
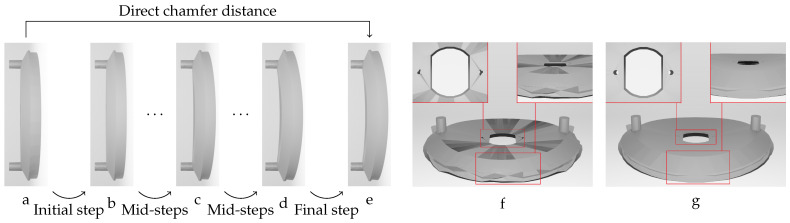
Sequential deformation (lower path) vs. direct application of the chamfer distance (upper path) on a vent cover object with significant geometric variation (**e**). (**a**) Initial undeformed model. (**b**) Least deformed model with MD=0.25 mm,F=10 N. (**c**) Average deformed model with MD=7.76 mm,F=300 N. (**d**) Deformed model with MD=14.75 mm,F=570 N. (**e**) Highest deformed model with MD=15.00 mm,F=580 N. Labels were generated using (**f**) direct chamfer distance and (**g**) sequential deformation algorithm.

**Figure 15 sensors-24-04707-f015:**
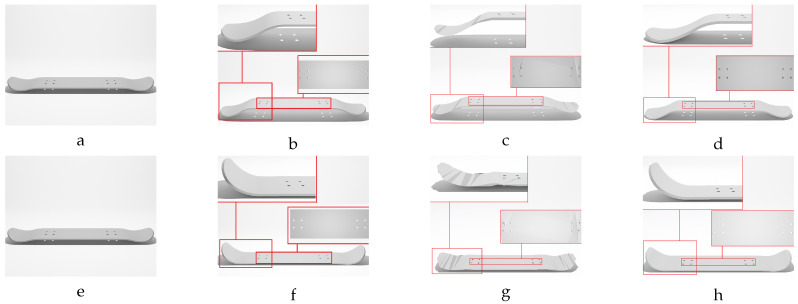
Sequential deformation (SD) vs. chamfer distance (CD) on skateboard object with MD=−19.93 mm (first row) and MD=19.93 mm (second row). (**a**,**e**) Initial undeformed model. (**b**,**f**) Deformed models (Ansys output). (**c**,**g**) CD algorithm outputs. (**d**,**h**) SD algorithm outputs.

**Figure 16 sensors-24-04707-f016:**
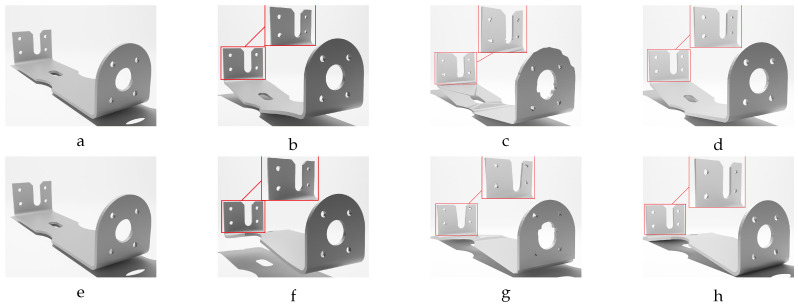
Sequential deformation (SD) vs. chamfer distance (CD) on bracket object with MD=−14.15 mm (first row) and MD=14.15 mm (second row). (**a**,**e**) Initial undeformed model. (**b**,**f**) Deformed models (Ansys output). (**c**,**g**) CD algorithm outputs. (**d**,**h**) SD algorithm outputs.

**Figure 17 sensors-24-04707-f017:**
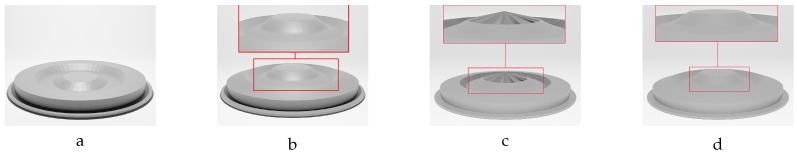
Sequential deformation (SD) vs. chamfer distance (CD) on RFRL object with MD=24.97 mm. (**a**) Initial undeformed model. (**b**) Deformed model (Ansys output). (**c**) CD algorithm output. (**d**) SD algorithm output.

**Figure 18 sensors-24-04707-f018:**
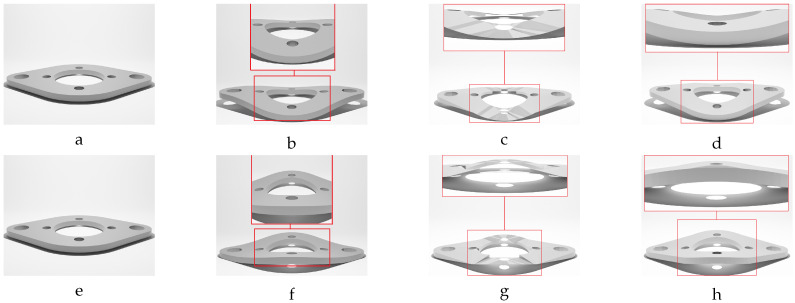
Sequential deformation (SD) vs. chamfer distance (CD) on oval flange object with MD=−18.20 mm (first row) and MD=18.20 mm (second row). (**a**,**e**) Initial undeformed model. (**b**,**f**) Deformed models (Ansys output). (**c**,**g**) CD algorithm outputs. (**d**,**h**) SD algorithm outputs.

**Figure 19 sensors-24-04707-f019:**
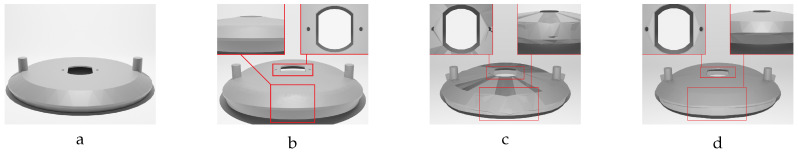
Sequential deformation (SD) vs. chamfer distance (CD) on vent cover object with MD=15.00 mm. (**a**) Initial undeformed model. (**b**) Deformed model (Ansys output). (**c**) CD algorithm output. (**d**) SD algorithm output.

**Figure 20 sensors-24-04707-f020:**
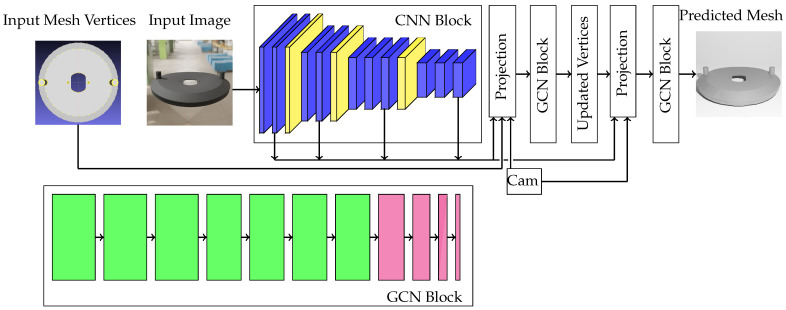
Training model pipeline: Blue cubes represent 2D convolutional layers, and yellow cubes denote max-pooling layers. The “cam” block represents the camera model’s intrinsic and extrinsic parameters. Green rectangles symbolize graph convolutional layers, while magenta rectangles represent dense layers.

**Figure 21 sensors-24-04707-f021:**
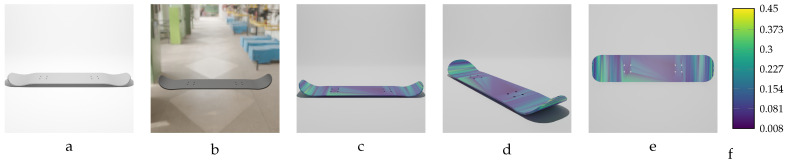
Visualization of actual Euclidean distance error between the predicted mesh and the ground truth for each vertex of the predicted skateboard [F=33.5 N,MD=15.0 mm]. (**a**) Input mesh of the training network (undeformed). (**b**) Input image of the deformed skateboard fed to the training model. (**c**–**e**) Predicted mesh of the deformed skateboard from different viewpoints. (**f**) Color bar representing the magnitude of the Euclidean distance error in millimeters (mm).

**Figure 22 sensors-24-04707-f022:**
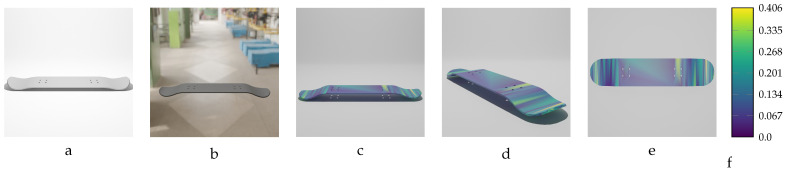
Visualization of actual Euclidean distance error between the predicted mesh and the ground truth for each vertex of the predicted skateboard [F=−33.5 N,MD=−15.0 mm]. (**a**) Input mesh of the training network (undeformed). (**b**) Input image of the deformed skateboard fed to the training model. (**c**–**e**) Predicted mesh of the deformed skateboard from different viewpoints. (**f**) Color bar representing the magnitude of the Euclidean distance error in millimeters (mm).

**Figure 23 sensors-24-04707-f023:**
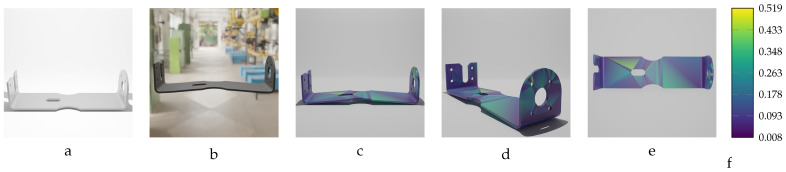
Visualization of actual Euclidean distance error between the predicted mesh and the ground truth for each vertex of the predicted bracket [F=28 N,MD=6.00 mm]. (**a**) Input mesh of the training network (undeformed). (**b**) Input image of the deformed bracket fed to the training model. (**c**–**e**) Predicted mesh of the deformed bracket from different viewpoints. (**f**) Color bar representing the magnitude of the Euclidean distance error in millimeters (mm).

**Figure 24 sensors-24-04707-f024:**
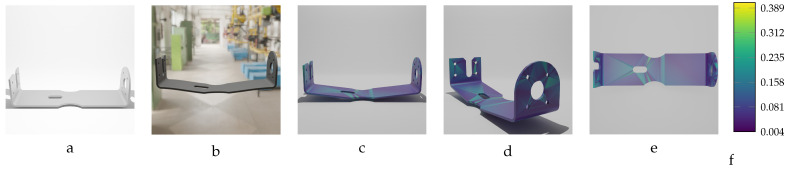
Visualization of actual Euclidean distance error between the predicted mesh and the ground truth for each vertex of the predicted bracket [F=−46 N,MD=−9.86 mm]. (**a**) Input mesh of the training network (undeformed). (**b**) Input image of the deformed bracket fed to the training model. (**c**–**e**) Predicted mesh of the deformed bracket from different viewpoints. (**f**) Color bar representing the magnitude of the Euclidean distance error in millimeters (mm).

**Figure 25 sensors-24-04707-f025:**
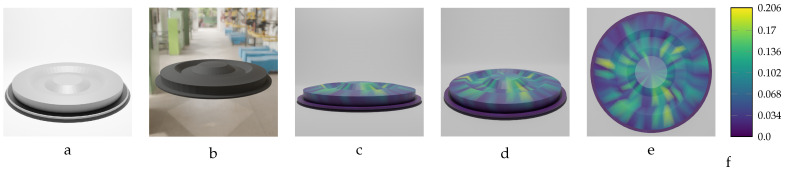
Visualization of actual Euclidean distance error between the predicted mesh and the ground truth for each vertex of the predicted RFRL [F=2750 N,MD=15.96 mm]. (**a**) Input mesh of the training network (undeformed). (**b**) Input image of the deformed RFRL fed to the training model. (**c**–**e**) Predicted mesh of the deformed RFRL from different viewpoints. (**f**) Color bar representing the magnitude of the Euclidean distance error in millimeters (mm).

**Figure 26 sensors-24-04707-f026:**
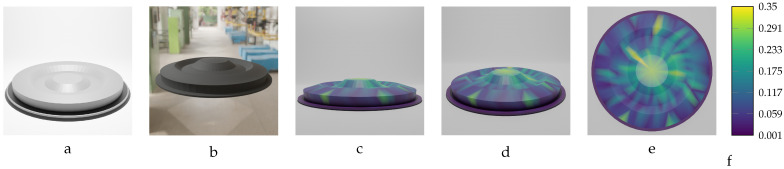
Visualization of actual Euclidean distance error between the predicted mesh and the ground truth for each vertex of the predicted RFRL [F=3950 N,MD=22.93 mm]. (**a**) Input mesh of the training network (undeformed). (**b**) Input image of the deformed RFRL fed to the training model. (**c**–**e**) Predicted mesh of the deformed RFRL from different viewpoints. (**f**) Color bar representing the magnitude of the Euclidean distance error in millimeters (mm).

**Figure 27 sensors-24-04707-f027:**
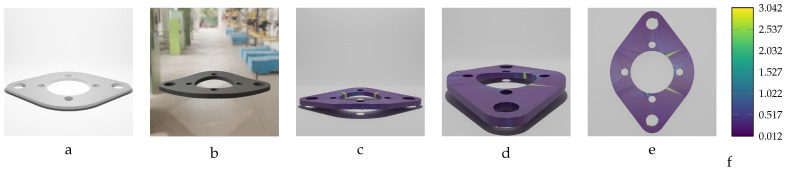
Visualization of actual Euclidean distance error between the predicted mesh and the ground truth for each vertex of the predicted oval flange [F=1750 N,MD=12.02 mm]. (**a**) Input mesh of the training network (undeformed). (**b**) Input image of the deformed oval flange fed to the training model. (**c**–**e**) Predicted mesh of the deformed oval flange from different viewpoints. (**f**) Color bar representing the magnitude of the Euclidean distance error in millimeters (mm).

**Figure 28 sensors-24-04707-f028:**
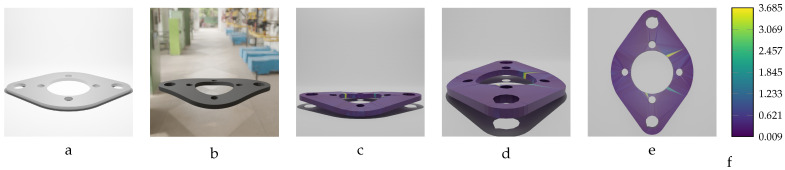
Visualization of actual Euclidean distance error between the predicted mesh and the ground truth for each vertex of the predicted oval flange [F=−2200 N,MD=−15.11 mm]. (**a**) Input mesh of the training network (undeformed). (**b**) Input image of the deformed oval flange fed to the training model. (**c**–**e**) Predicted mesh of the deformed oval flange from different viewpoints. (**f**) Color bar representing the magnitude of the Euclidean distance error in millimeters (mm).

**Figure 29 sensors-24-04707-f029:**
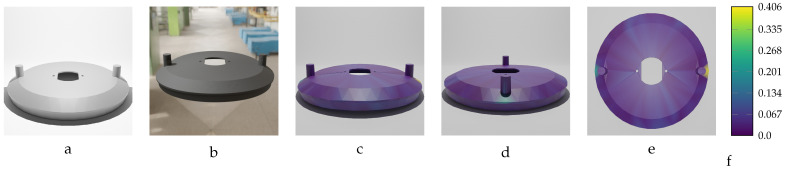
Visualization of actual Euclidean distance error between the predicted mesh and the ground truth for each vertex of the predicted vent cover [F=310 N,MD=8.02 mm]. (**a**) Input mesh of the training network (undeformed). (**b**) Input image of the deformed vent cover fed to the training model. (**c**–**e**) Predicted mesh of the deformed vent cover from different viewpoints. (**f**) Color bar representing the magnitude of the Euclidean distance error in millimeters (mm).

**Figure 30 sensors-24-04707-f030:**
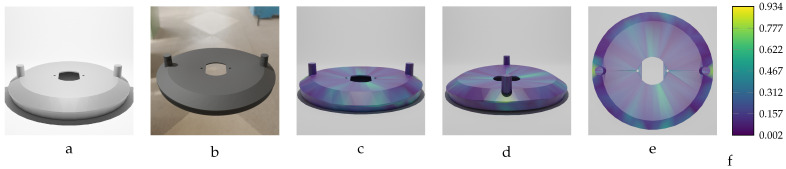
Visualization of actual Euclidean distance error between the predicted mesh and the ground truth for each vertex of the predicted vent cover [F=390 N,MD=−10.09 mm]. (**a**) Input mesh of the training network (undeformed). (**b**) Input image of the deformed vent cover fed to the training model. (**c**–**e**) Predicted mesh of the deformed vent cover from different viewpoints. (**f**) Color bar representing the magnitude of the Euclidean distance error in millimeters (mm).

**Figure 31 sensors-24-04707-f031:**
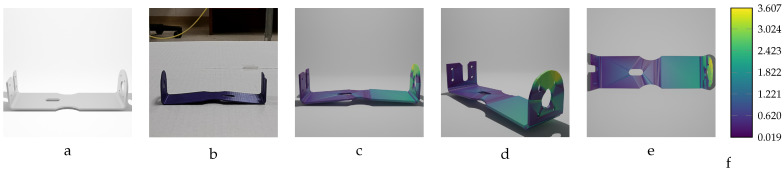
Visualization of actual Euclidean distance error between the predicted mesh and the ground truth for each vertex of the predicted bracket [F=−28 N,MD=−6.00 mm]. (**a**) Input mesh of the training network (undeformed). (**b**) Real image of the actual deformed bracket fed to the training model. (**c**–**e**) Predicted mesh of the deformed bracket from different viewpoints. (**f**) Color bar representing the magnitude of the Euclidean distance error in millimeters (mm).

**Figure 32 sensors-24-04707-f032:**
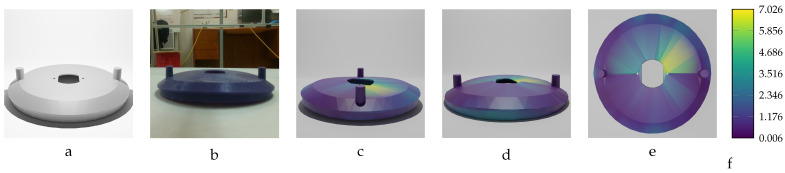
Visualization of actual Euclidean distance error between the predicted mesh and the ground truth for each vertex of the predicted vent cover [F=500 N,MD=−10.09 mm]. (**a**) Input mesh of the training network (undeformed). (**b**) Real image of the 3D-printed deformed vent cover fed to the training model. (**c**–**e**) Predicted mesh of the deformed vent cover from different viewpoints. (**f**) Color bar representing the magnitude of the Euclidean distance error in millimeters (mm).

**Table 1 sensors-24-04707-t001:** Number of vertices and faces of studied undeformed objects.

Object	Number of Vertices	Number of Faces	Related Figure
Skateboard	443	914	Figure 1a
Bracket	756	1548	Figure 1b
RFRL ${\;}^{a}$	706	1408	Figure 1c
Oval Flange	860	1744	Figure 1d
Vent Cover	1145	2298	Figure 1e

${\;}^{a}$ Round flat receptacle lid.

**Table 2 sensors-24-04707-t002:** Range and step of applied forces and resulting deformations.

Object	Force Range (N)	Force Step (N)	MD ^a^ Range (mm)	MD Step (mm)
Skateboard	−44.5 to 44.5	0.5	−19.93 to 19.93	0.22
Bracket	−66.0 to 66.0	2.0	−14.15 to 14.15	0.43
RFRL ${\;}^{a}$	50 to 4300	50.0	0.29 to 24.97	0.29
Oval Flange	−2650 to 2650	50.0	−18.20 to 18.20	0.34
Vent Cover	−580 to 580	10.0	−15.00 to 15.00	0.25

${\;}^{a}$ Maximum deformation (MD) at applied force.

**Table 3 sensors-24-04707-t003:** Details of scene and camera configuration.

Property	Setting
Light Type	Point light, 200 mW
Render Engine	Cycles
Camera Sensor Size	36mm
Lens Type	Perspective
Focal Length	58.0 m m
Image Resolution	600×600 pixels
Background	HDRI ${\;}^{a}$: Boiler_room_1K(CC0 ${\;}^{b}$ license)

${\;}^{a}$ HDRI: high-dynamic-range image. ${\;}^{b}$ CC0: no rights reserved.

**Table 4 sensors-24-04707-t004:** Camera, light, and object positions/rotations for rendering.

Object	$R_{Z_{O}}{(}^{\circ })$	$T_{X_{L}}\left(\mathrm{mm}\right)$	$R_{X_{C}}{(}^{\circ })$	Number of Images
Skateboard	{−60:15:90}	{5:20:45}	{2:5:27}	180
Bracket	{0:30:330}−{270}	{0:15:45}	{0:10:50}	240
RFRL	{0}	{0:5:50}	{0:5:50}	121
Oval Flange	{0:10:90}	{5:20:45}	{0:5:25}	180
Vent Cover	{0:30:150}	{25:10:45}+{0}	{0:10:60}	168

**Table 5 sensors-24-04707-t005:** Intel RealSense camera calibration results.

Image Size	RDC ^a^	Skew	TDC ^b^	Focal Length	Principal Point
1280×720	[0.1731,−0.5222,0.4402]	−0.9725	[0.0012,−0.001]	[907.7907,907.9046]	[630.1808,371.5688]

${\;}^{a}$ RDC: radial distortion coefficient. ${\;}^{b}$ TDC: tangential distortion coefficient.

**Table 6 sensors-24-04707-t006:** Training hyperparameters and execution time per object’s model.

Object	Batch ^a^	Train Data	Test Data	Valid ^b^ Data	Train T ^c^ (h)	Test T ^d^ (h,s)	Epochs
Skateboard	15	2640	990	330	38.38	0.35,1.27	47
Bracket	15	2880	1200	720	55.5	0.52,1.56	45
RFRL	11	1680	500	240	27.41	0.2,1.44	35
Oval Flange	12	2340	900	360	47.04	0.38,1.52	47
Vent Cover	11	2398	968	330	56.11	0.45,1.67	48

${\;}^{a}$ Batch size used during training each model. ${\;}^{b}$ Validation. ${\;}^{c}$ Time in hours. ${\;}^{d}$ Time for entire test data (hours, first number), and per instance (seconds, second number).

**Table 7 sensors-24-04707-t007:** Performance of the trained model on the studied objects.

Object	$MAcc^{★}$ ^a^ (mm)	$SdevAcc^{★}$ ^b^ (mm)
Skateboard	0.001461	0.000276
Bracket	0.001595	0.000241
RFRL	0.000699	0.000263
Oval Flange	0.002403	0.000818
Vent Cover	0.002226	0.000212

${\;}^{a}$ Normalized mean Euclidean error. ${\;}^{b}$ Standard deviation of the normalized Euclidean error.

**Table 8 sensors-24-04707-t008:** Evaluation of the training model performance on the skateboard.

Applied Force (N)	MD (mm)	$MAcc^{★}$ (mm)	$SdevAcc^{★}$ (mm)	MAcc (mm)	SdevAcc (mm)
22.5	10.08	0.001816	0.000282	0.240	0.037
25.0	11.20	0.001476	0.000200	0.195	0.026
27.0	12.09	0.001333	0.000196	0.176	0.025
29.0	12.99	0.001288	0.000184	0.170	0.024
31.5	14.11	0.001200	0.000187	0.158	0.024
33.5	15.00	0.001194	0.000214	0.158	0.028
35.5	15.90	0.001230	0.000181	0.162	0.024
38.0	17.02	0.001312	0.000242	0.173	0.032
40.0	17.92	0.001389	0.000184	0.183	0.024
42.5	19.04	0.001658	0.000282	0.219	0.037
44.5	19.93	0.001996	0.000395	0.264	0.052
−22.5	−10.08	0.001732	0.000376	0.229	0.049
−25.0	−11.20	0.001374	0.000258	0.181	0.034
−27.0	−12.09	0.001340	0.000251	0.177	0.033
−29.0	−12.99	0.001241	0.000208	0.164	0.027
−31.5	−14.11	0.001243	0.000198	0.164	0.026
−33.5	−15.00	0.001221	0.000223	0.161	0.029
−35.5	−15.90	0.001275	0.000211	0.168	0.027
−38.0	−17.02	0.001352	0.000219	0.178	0.028
−40.0	−17.92	0.001534	0.000273	0.202	0.036
−42.5	−19.04	0.001811	0.000499	0.239	0.066
−44.5	−19.93	0.002138	0.000816	0.282	0.107

**Table 9 sensors-24-04707-t009:** Evaluation of the training model performance on the bracket.

Applied Force (N)	MD (mm)	$MAcc^{★}$ (mm)	$SdevAcc^{★}$ (mm)	MAcc (mm)	SdevAcc (mm)
24	5.14	0.001571	0.000263	0.157	0.026
28	6.00	0.001405	0.000209	0.135	0.019
32	6.86	0.001421	0.000208	0.142	0.020
38	8.15	0.001484	0.000228	0.148	0.022
42	9.00	0.001535	0.000223	0.153	0.022
46	9.86	0.001564	0.000236	0.156	0.023
52	11.15	0.001637	0.000229	0.163	0.022
56	12.01	0.001688	0.000238	0.168	0.023
60	12.87	0.001896	0.000416	0.189	0.041
66	14.15	0.002449	0.000508	0.244	0.050
−24	−5.14	0.001436	0.000195	0.143	0.019
−28	−6.00	0.001358	0.000191	0.135	0.019
−32	−6.86	0.001352	0.000194	0.135	0.019
−38	−8.15	0.001425	0.000177	0.142	0.017
−42	−9.00	0.001388	0.000172	0.138	0.017
−46	−9.86	0.001419	0.000182	0.141	0.018
−52	−11.15	0.001510	0.000215	0.151	0.021
−56	−12.01	0.001595	0.000242	0.159	0.024
−60	−12.87	0.001701	0.000232	0.170	0.023
−66	−14.15	0.002074	0.000264	0.207	0.026

**Table 10 sensors-24-04707-t010:** Evaluation of training model performance on the round flat receptacle lid (RFRL).

Applied Force (N)	MD (mm)	$MAcc^{★}$ (mm)	$SdevAcc^{★}$ (mm)	MAcc (mm)	SdevAcc (mm)
1050	6.09	0.000746	0.000309	0.093	0.038
1200	6.96	0.000600	0.000296	0.075	0.037
1400	8.12	0.000615	0.000196	0.076	0.024
1550	9.00	0.000645	0.000230	0.080	0.028
1700	9.87	0.000521	0.000115	0.065	0.014
1900	11.03	0.000600	0.000192	0.075	0.024
2050	11.90	0.000656	0.000221	0.082	0.027
2250	13.06	0.000663	0.000283	0.083	0.035
2400	13.93	0.000660	0.000184	0.082	0.023
2600	15.09	0.000587	0.000219	0.073	0.027
2750	15.96	0.000585	0.000163	0.073	0.020
2950	17.13	0.000681	0.000297	0.085	0.037
3100	18.02	0.000664	0.000183	0.083	0.022
3250	18.87	0.000802	0.000448	0.100	0.056
3450	20.03	0.000723	0.000291	0.090	0.036
3600	20.90	0.000718	0.000204	0.089	0.025
3800	22.06	0.000841	0.000342	0.105	0.042
3950	22.93	0.000801	0.000196	0.100	0.024
4150	24.09	0.000919	0.000451	0.114	0.056
4300	24.97	0.000964	0.000441	0.120	0.055

**Table 11 sensors-24-04707-t011:** Evaluation of the training model performance on the oval flange.

Applied Force (N)	MD (mm)	$MAcc^{★}$ (mm)	$SdevAcc^{★}$ (mm)	MAcc (mm)	SdevAcc (mm)
1300	8.93	0.002474	0.000720	0.395	0.115
1450	9.96	0.002497	0.000848	0.399	0.135
1600	10.99	0.002393	0.000834	0.382	0.133
1750	12.02	0.002246	0.000761	0.359	0.121
1900	13.05	0.002282	0.000796	0.365	0.127
2050	14.08	0.002435	0.000915	0.389	0.146
2200	15.11	0.002386	0.000814	0.381	0.130
2350	16.14	0.002488	0.000901	0.398	0.144
2500	17.17	0.002679	0.000846	0.428	0.135
2650	18.20	0.003055	0.000970	0.488	0.155
−1300	−8.93	0.002285	0.000662	0.365	0.106
−1450	−9.96	0.002312	0.000783	0.369	0.125
−1600	−10.99	0.002074	0.000628	0.331	0.100
−1750	−12.02	0.002151	0.000709	0.344	0.113
−1900	−13.05	0.002096	0.000753	0.335	0.120
−2050	−14.08	0.002208	0.000833	0.353	0.133
−2200	−15.11	0.002193	0.000775	0.351	0.124
−2350	−16.14	0.002372	0.000838	0.379	0.134
−2500	−17.17	0.002492	0.000893	0.398	0.142
−2650	−18.20	0.002947	0.001100	0.471	0.176

**Table 12 sensors-24-04707-t012:** Evaluation of the training model performance on the vent cover.

Applied Force (N)	MD (mm)	$MAcc^{★}$ (mm)	$SdevAcc^{★}$ (mm)	MAcc (mm)	SdevAcc (mm)
540	13.97	0.003919	0.000326	0.479	0.039
500	12.93	0.003147	0.000269	0.384	0.032
460	11.90	0.002679	0.000203	0.327	0.024
430	11.12	0.002444	0.000232	0.298	0.028
390	10.09	0.002093	0.000194	0.255	0.023
350	9.05	0.001889	0.000227	0.231	0.027
310	8.02	0.001624	0.000180	0.198	0.022
270	6.98	0.001718	0.000182	0.210	0.022
230	5.95	0.001912	0.000145	0.233	0.017
190	4.91	0.002184	0.000163	0.266	0.020
150	3.88	0.002768	0.000233	0.338	0.028
−540	−13.97	0.002737	0.000390	0.334	0.047
−500	−12.93	0.002178	0.000242	0.266	0.029
−460	−11.90	0.001832	0.000189	0.224	0.023
−430	−11.12	0.001664	0.000190	0.203	0.023
−390	−10.09	0.001565	0.000210	0.191	0.025
−350	−9.05	0.001585	0.000189	0.193	0.023
−310	−8.02	0.001704	0.000173	0.208	0.021
−270	−6.98	0.001870	0.000160	0.228	0.019
−230	−5.95	0.002149	0.000192	0.262	0.023
−190	−4.91	0.002417	0.000177	0.295	0.021
−150	−3.88	0.002904	0.000201	0.354	0.024

**Table 13 sensors-24-04707-t013:** Evaluation of the training model performance on the real bracket data.

Applied Force (N)	MD (mm)	$MAcc^{★}$ (mm)	$SdevAcc^{★}$ (mm)	MAcc (mm)	SdevAcc (mm)
28	6.00	0.011240	0.001713	1.080	0.155
38	8.15	0.012168	0.001778	1.213	0.1716
46	9.86	0.013450	0.002006	1.341	0.195
56	12.01	0.013335	0.001832	1.327	0.177
66	14.15	0.022041	0.004572	2.220	0.450
−28	−6.00	0.009913	0.001337	0.985	0.133
−38	−8.15	0.011115	0.001342	1.107	0.129
−46	−9.86	0.012061	0.001547	1.198	0.153
−56	−12.01	0.015631	0.002347	1.558	0.232
−66	−14.15	0.023228	0.002613	2.318	0.257

**Table 14 sensors-24-04707-t014:** Evaluation of the model performance on the real vent cover data.

Applied Force (N)	MD (mm)	$MAcc^{★}$ (mm)	$SdevAcc^{★}$ (mm)	MAcc (mm)	SdevAcc (mm)
500	12.93	0.012179	0.003882	1.488	0.474
430	11.12	0.023498	0.008127	2.871	0.993
350	9.05	0.015358	0.003762	1.877	0.459
270	6.98	0.019515	0.002918	2.385	0.356
190	4.91	0.014924	0.004478	1.824	0.547
−500	−12.93	0.023948	0.008013	2.926	0.979
−430	−11.12	0.015752	0.006917	1.925	0.845
−350	−9.05	0.019591	0.007357	2.394	0.899
−270	−6.98	0.013490	0.005645	1.649	0.690
−190	−4.91	0.014701	0.004158	1.797	0.508

## Data Availability

A limited portion of the dataset is currently available for public access on GitHub https://github.com/Sahand-adli/Deformation-Estimation-Single-Image (accessed on 17 April 2024). Due to restrictions associated with free GitHub accounts, only a subset of the data is uploaded at this time. The code used in this work will be made available upon paper acceptance for publication.

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
