# Peer review of "Deformation Estimation of Textureless Objects from a Single Image"

_sensors, 2024, doi:10.3390/s24144707_

Round 1

Reviewer 1 Report

Comments and Suggestions for Authors

The article proposes a method for estimating the deformation of texture-less plastic objects using a single RGB image. The key components of the method are: (1) Creation of a unique image dataset containing five deformed parts. Then,  the dataset includes initial undeformed models, various marginally deformed versions, and corresponding RGB images from different viewpoints. (2) It conducuts a new method for generating mesh labels to represent deformations. (3) It designes graph convolution training model to project object vertices into features extracted from the input image. (4) It addresses the challenge of mapping 2D image features to 3D deformations.

Advantages:

(1) It achieves sub-millimeter accuracy for synthetic images and around 2.0 mm for real images, demonstrating high precision in deformation estimation.

(2) Sequential deformation method offers a more precise alternative to existing chamfer distance algorithms for mesh label generation.

(3) It focuses on using a single RGB image aligns with practical industrial inspection scenarios, making the method highly applicable.

(4) It utilizes advanced graph convolution techniques to map 2D image features to 3D deformations effectively.

Disadvantages:

(1) The study's reliance on a specific dataset of five deformed parts may limit the generalizability of the results to a broader range of objects and deformations.

(2) Using a single image may not capture all aspects of complex deformations that could be observed from multiple viewpoints.

(3) The method is tailored to plastic objects, potentially limiting its applicability to other materials with different deformation characteristics.

(4) The use of advanced graph convolution models may introduce significant computational complexity, affecting real-time applicability.

(5) The method requires an initial undeformed CAD model, which may not always be available or practical in all industrial scenarios.

Comments on the Quality of English Language

It seems to be good.

Reviewer 2 Report

Comments and Suggestions for Authors

The manuscript is devoted to the problem of object deformation estimation from a single image. The authors propose a solution for textureless plastic objects. They also developed an original dataset of images designed to train the network. I consider the results of the study presented in the manuscript to be good and interesting. Overall, the manuscript is well organized, written in good English and looks convincing. I have the following remarks to the text.

1) The Introduction looks long. I would recommend to shorten it about twice.

2) The manuscript does not provide any data on the computational time and efficiency of the proposed methods and network training. This information should be added.

3) It would be better to avoid a bulk reference like “[13-18]” in line 140. These references (except [14]) are also not analyzed further. I would recommend either providing an analysis of referenced articles or shortening this reference to a couple of the most important articles.

Also, taking into account that technologies for reconstructing objects from a single image are important for this study, and they are developing rapidly, IMO recent works (for example, https://doi.org/10.48550/arXiv.2312.09147 and  https://doi.org/10.3390/sym16020184 published half a year ago) will be of interest to the authors.

4) Line 463. “Both SD (third column) and CD (fourth column) algorithms” contradicts Figures 15-19 where CD algorithm is represented by third column and SD algorithm – by fourth one.

5) In Fig. 16(h) wrongly depicts the skateboard object instead of the bracket.
